Progress in overactive bladder: novel avenues from psychology to clinical opinions

Jin Zhaofeng 1
Zhang Qiumin 1
Yu Yanlan 2
Zhang Ruilin 1
Ding Guoqing 2
Li Tian fmmult@foxmail.com 3
Song Yuping ypsong1994@126.com 1
1 School of Psychology, Weifang Medical University , Weifang , China
2 Department of Urology, Sir Run Run Shaw Hospital, School of Medicine, Zhejiang University , Hangzhou , China
3 School of Basic Medicine, Fouth Military Medical University , Xi’an , China
Qin Jiangjiang
Electronic publication date: 2023 Nov 1
Publication date: 2023
Volume: 11
Electronic Location ID: e16112
Received 2023 Mar 7; Accepted 2023 Aug 27
Copyright: ©2023 Jin et al.
Copyright year: 2023
Copyright holder: Jin et al.
License: This is an open access article distributed under the terms of the Creative Commons Attribution License, which permits unrestricted use, distribution, reproduction and adaptation in any medium and for any purpose provided that it is properly attributed. For attribution, the original author(s), title, publication source (PeerJ) and either DOI or URL of the article must be cited.
License URL: https://creativecommons.org/licenses/by/4.0/

Keywords: Overactive bladder, Mental health, Social-psychological factors, Psychological intervention

Funding: Zhejiang Provincial Key Research and Development Program 2021C03062 This article is supported by the Zhejiang Provincial Key Research and Development Program (2021C03062). The funders had no role in study design, data collection and analysis, decision to publish, or preparation of the manuscript.

==============================
Rationale

Overactive bladder (OAB) is a common, distressing condition that worsens with age and impacts quality of life significantly. As a results of its clinical symptoms, patients suffer from serious physical and mental health issues, have a poor quality of life, and participate in a serious economic burden. The key social-psychological factors include living habits, eating habits, and personality characteristics on this disease, even though the pathogenesis of OAB is complex. However, there is few cognitions and research on OAB in the field of psychology.

Methods/Search Strategy

Between 2000 and 2022, two electronic databases were systematically searched in accordance with Cochrane library guidelines (PubMed/Medline, Web of Science). An analysis of the remaining articles with relevant information was conducted using a data extraction sheet. An itemized flow diagram was adopted and used to report systematic reviews and meta-analysis. A systematic review of studies published from 2000 to 2022 in English language were conducted and included in the review.

The intended audience

Urological surgeon and psychologists majoring in urinary diseases.

Implication

As a result of this information, we are able to develop a better understanding of the role of psychological factors in the development of OAB and suggest potential therapeutic directions for OAB patients. This may benefit the recovery of OAB patients.

Introduction

Overactive bladder (OAB) is a condition affecting millions of individuals in the United States. It is a syndrome characterized by urgency of urination, often with frequency of urination and increased nocturia, with or without urge incontinence, no urinary tract infection or other significant pathological changes (Paul et al., 2003). The incidence rate of OAB ranges from 2.1% to 30.9% (Irwin et al., 2006; Sellers & Donna, 2007; Ikeda & Nozoawa, 2015; Wen et al., 2014; Wang et al., 2011), which ranks among the top ten of all chronic diseases and increases with the increase of age. OAB has become an important public health problem threatening global human health (Sellers & Donna, 2007; Wang et al., 2011; Wyman, Burgio & Newman, 2009). As a result of OAB, patients suffer from physiological, psychological, social, and sexual issues (Kinsey et al., 2016; Sexton et al., 2011). On the contrary, the long-term existence of the problems above may aggravate OAB patients and result in reduced quality of life (QOL), resulting in a vicious cycle (Evans et al., 2005).

Compared with drug therapy alone, adding psychological intervention to drug therapy for OAB patients could achieve better results, the study on 111 female OAB patients which conducted by Yu et al. (2017) found. With the development of bio-psycho-social medical model (Wang, Shi & Lu, 2019) and the proposal of precision medicine model (König et al., 2017), the field of psychosomatic medicine has entered a booming era: More attention has been paid to the mental health status and social-psychological factors of patients. How to use psychological knowledge and psychological means to intervene in the course of disease has become an urgent issue to be solved.

This review focuses on the psychological research progress of OAB from the perspective of psychology and clinical opinions. First, we introduce the epidemiological characteristics of overactive bladder. Thereafter, we highlight the psychological etiology and the psychological intervention of OAB. In summary, the information compiled here will provide a strong reference for the treatment of OAB and is expected to be helpful for the rehabilitation of OAB patients.

Epidemiological characteristics of OAB

OAB can be divided into dry OAB and wet OAB (Paul et al., 2003) depending on whether it is accompanied by symptoms of acute urinary incontinence. The probability of dry OAB versus wet OAB is about 2 to 1 (Chen et al., 2012; Anger et al., 2012; Evans et al., 2005; Yu et al., 2017). Among them, male patients are more likely to have dry OAB, while female patients have wet OAB, which may be related to women’s relatively weak bladder necks and urethral sphincters (Stewart et al., 2003), and may also be related to the increased frequency of urination due to the small bladder volume in female patients (Hsiao et al., 2019). Based on large epidemiological surveys at home and abroad, OAB prevalence ranges from 2.1% to 30.9% (Irwin et al., 2006; Sellers & Donna, 2007; Ikeda & Nozoawa, 2015; Wang et al., 2011), and increases with age (Sellers & Donna, 2007; Wang et al., 2011). More OAB symptoms as nocturia, urgency of urination and frequency of urination have been reported by females at each period of time (Irwin et al., 2006). Different definitions of OAB, different methods of investigation and research, and regional differences may be the reasons for such differences (Wen et al., 2014). At present, the number of OAB patients ranks in the top ten of all chronic diseases and increases with aging, which brings serious financial burden to families of OAB patients and health insurance funds (Wang et al., 2011). OAB itself is not an indication of hospitalization, many patients are hospitalized according to their condition when most patients are treated and followed up on an outpatient basis (Firoozi et al., 2013). OAB has become an important global public health problem and a threat to human health (Wyman, Burgio & Newman, 2009).

However, in sharp contrast to the high incidence rate is the low rate of medical care, meaning that most patients do not choose to seek care (Ricci et al., 2001). Foreign studies have shown that only 38% to 60% of OAB patients have consulted medical staff about their urination symptoms (Milsom et al., 2001; Irwin et al., 2008). An analysis of lower urinary tract symptoms among China’s first outpatient patients with urinary surgery (LUTS China) showed that 85% of OAB patients had not sought medical treatment, and 59% of patients had moderate to severe symptoms when seeking medical treatment. Among them, the low awareness of the disease is an important reason for the low rate of medical treatment. Patients believe that OAB-related symptoms such as urgency, frequency of urination and nocturia will occur naturally as they get older, preventing them from seeking medical help, not knowing that this is a treatable disease (Yanhong, 2016). Others delay treatment because they are shy or unable to seek professional help (Iyer et al., 2021).

In addition, OAB is a long-term chronic disease characterized by a high recurrence rate. Kim et al. (2017) conducted a study on 441 patients with OAB who were effective in drug therapy, and found that among 371 patients with OAB who completed 6-month follow-up, the cumulative recurrence rates at 1, 3 and 6 months after drug withdrawal were 25.6%, 42.3% and 52.2%, respectively. Currently, the treatment options for OAB patients are behavioral therapy (Liu, Jin & Zhang, 2019) and drug therapy (Kreydin, Gomes & Cruz, 2021), while other options include intravesical drug infusion (Qin, Gao & Xia, 2019; Fang, Wu & Bixia, 2021), botox type A detrusor injection (Wells et al., 2020), sacral nerve conditioning (Reekmans et al., 2021), and surgical treatment (Juszczak et al., 2019), etc. Compared to other treatments, the cycle of bladder training and other behavioral treatments is longer, and the patients’ compliance is poorer (Liu, Jin & Zhang, 2019). The overall effective rate of drug treatment is not high, while there are a number of side effects, including dry mouth, constipation, nausea and urinary retention (Kreydin, Gomes & Cruz, 2021; Qin, Gao & Xia, 2019; Fang, Wu & Bixia, 2021). Both surgery and sacral nerve conditioning are invasive and expensive (Reekmans et al., 2021; Juszczak et al., 2019). The long-term and repetitive nature of OAB treatment has caused serious trouble to patients’ daily life.

Research progress in psychological etiology of OAB

The influencing factors of OAB are many and complex. With the proposal and development of the bio-psycho-social medical model, people gradually realize that it is not complete to explain the pathogenesis of OAB solely from the pathophysiological perspective of the disease. The impact of the disease on the quality of life and psychological level of patients should be considered when curing it.

In the past, the main direction of conquering this disease was the development of new drugs and innovative therapeutic methods, such as micro-radiofrequency therapy (Okhunov et al., 2019), etc. However, due to the characteristics of high incidence, low hospitalization rate, high recurrence rate and no organic lesions, the prognosis of OAB is not significantly improved by current therapeutic methods (Liu, Jin & Zhang, 2019; Kreydin, Gomes & Cruz, 2021). For better therapeutic outcomes, behavioral therapy should be combined with psychological intervention as the first-line treatment of OAB (Yu et al., 2017). Psychosocial factors can affect the mental and psychological status of OAB patients, so the study of psychosocial factors in the occurrence and development of OAB should be paid attention to (Kinsey et al., 2016; Sexton et al., 2011; Evans et al., 2005).

Personality characteristics

A large amount of studies have shown that emotions are closely related to people’s health, and long-term negative emotional state in OAB patients will damage their physical health, in which personality traits such as mental resilience (Chen et al., 2016; Fan, Meng & Ping, 2015), self-efficacy (Li et al., 2018) and self-esteem (Kinsey et al., 2016) play an important role (Wang et al., 2011).

Psychological resilience refers to an individual’s ability to adapt to life adversity, trauma, tragedy, threat or other major life pressure (Chen et al., 2016). It is a new concept and research hotspot in the field of psychology to look at stress response and adaptability about individual from a new perspective (Zebhauser et al., 2014). Through a questionnaire survey of 80 female patients with refractory OAB, Chen et al. (2016) found that those with higher mental resilience tended to have extroverted personality and high degree of social support. And low level of mental resilience often physical symptoms are more serious, neurotic, accompanied by anxiety and depression.

Self-efficacy is a person’s guess and assessment of whether he or she is capable enough to perform certain actions (Lei et al., 2021). The treatment of clinical diseases requires patients’ cooperation, while the initiative and compliance of patients in the treatment of functional diseases become the key factors to obtain significant curative effect due to the absence of organic lesions and the fact that the treatment plan is often combined with psychological intervention (Yu et al., 2017). He et al. (2017) found that self-efficacy theory can improve patients’ self-efficacy and compliance of medical treatment so as to consolidate treatment effect. Li et al. (2018) found that self-efficacy intervention could effectively improve the psychological status and life quality of OAB patients.

Self-esteem is a comprehensive evaluation of the degree of respect that an individual makes on the basis of self-image and self-value, and it is the degree of recognition of a person’s sense of value of life (Xu et al., 2022). Dmochowski & Newman (2007) found that patients with OAB reported higher levels of low self-esteem than participants without the condition. On this basis, Nicolson et al. (2008) related low self-esteem with poor physical condition and studied its role in OAB disease progression. Snow-Lisy (2018) has found that children with urgent and frequent urination suffer poor self-esteem and quality of life.

Coping style

The way individuals cope with the disease also has an important effect on the mental health of OAB patients (Ricci et al., 2001; Wang et al., 2011). Coping style is represented by the cognitive and behavioral styles that individuals adopt when facing OAB, which are determined by different personality types. Different coping styles affect the physical and mental health of patients to different degrees. As an intermediary factor, coping style affects the nature and intensity of stress response and regulates the dynamic balance between stress and OAB. Among patients with urinary incontinence, it was found that the negative coping style significantly correlated with the progression of urinary incontinence and the reduction of quality of life (Li et al., 2014).

The way patients cope with OAB affects their emotional state. Blasco et al. (2017) found that many elderly OAB patients regard OAB symptoms as common symptoms caused by aging and do not seek medical treatment, and some even choose to go to the hospital until they have serious symptoms such as urinary incontinence. As a result, OAB patients suffer from OAB symptoms for a long time and show emotions like depression, anxiety and so on. It is more common for OAB patients to use negative coping styles such as avoidance when they are depressed. The severity of symptoms in OAB patients is obviously affected by mood, which can aggravate the clinical symptoms of patients.

Cognition

Correct understanding of the disease is the prerequisite of treatment. Study found that not only did patients not have a good understanding of OAB (Yanhong, 2016), but even non-urologists did not have a good understanding of OAB-related treatment options (Guo & Yang , 2020). The low awareness of OAB among patients and some medical staff contributes to the high incidence and low hospitalization rate of OAB. And we have a long way to go to change that:

(1) Hold a series of academic conferences and professional lectures for medical staff to improve their awareness of OAB diagnosis and treatment protocols.

(2) Strengthen outpatient health education to ensure that OAB patients can fully understand the disease and cooperate with doctors in diagnosis and treatment.

(3) Actively do a good job in the scientific popularization of disease-related knowledge, improve the public’s understanding of the disease, and strive to improve the rate of the disease.

Other psychological etiology

Various reasons acting on the body can cause OAB and lead to psychological problems, and psychological problems can in turn aggravate OAB symptoms, further reduce the patients’ quality of life, so repeatedly forming a vicious cycle (Kinsey et al., 2016; Sexton et al., 2011; Evans et al., 2005). According to the psychopathological network theory, a stressor acting on the body will make it in a continuous negative emotional state and cause multiple symptoms, and the interaction between symptoms can further aggravate the negative emotions, so that even if the initial stressor is removed, these symptoms can continue to interact with each other and maintain the disease state (Chen et al., 2021).

We can also find from previous studies that the occurrence and development of OAB may be affected by individual age, growth environment, nature of work, pressure, holding urine, urination without intention, etc. For example, the prevalence rate of female nurses in Grade A tertiary hospital is 19.2%, much higher than that of ordinary Chinese women, which is 6%∼8% (Wang et al., 2011). This may be closely related to the busy work of nurses in Grade A tertiary hospitals, irregular work and rest times and unhealthy urination behaviors, such as holding urine, caused by their special nature of work (Chen et al., 2015).

Methods

Search strategy

Cochrane library guidelines were followed when conducting this systematic review. Studies were accessed through Pubmed and Web of Science using terms applicable to specific databases. Additionally, “Booleans” operator (AND and OR) were also used to acquire best information between the following keywords: “overactive,” “OAB,” “Detrusor hyperactivity,” “unstable bladder,” psychotherapy, health education and quality of life. We used these to find relevant studies that explore the factors affecting OAB patients’ health outcomes.

Selection criteria

We extracted relevant articles concerning OAB patients in hospitals and other health facilities from all types of studies (qualitative, quantitative and mixed- method) worldwide. Studies published in peer-reviewed journals, published books and WHO reports with full text available that were related to quality of life of OAB patients between 1 January 2000 and 31 December 2022 and in English language were evaluated and included. The exclusion criteria included systematic review studies and studies with no full text available.

Selection process

Identifying pertinent articles begins with checking all titles and abstracts. After the abstracts, the full texts of all remaining articles were reviewed to determine if they are relevant to this study. We also checked for duplicate articles or studies, and only articles that met the inclusion and exclusion criteria were saved for later use. Furthermore, the bibliography of all the remaining studies was checked to find some other publications not included in the selected databases. A total of 21 articles were retrieved, addressing the factors that affect the quality of life of patients with OAB. The impact of OAB on the physiological and psychological well-being of patients and the psychological interventions that can improve their quality of life. A major theme was then assigned to each of the selected articles. The Preferred Reporting Items for Systematic Reviews and Meta-Analysis (PRISMA) Flow Diagram in Fig. 1 was adopted and used as a preferred reporting item (Moher et al., 2009).

Figure 1 Article search and selection process.

A preferred reporting items for systematic reviews and meta- analysis (PRISMA) flow diagram in Fig. 1 was adopted and used as a preferred reporting item for systematic reviews.

This study was conducted with the help of a data extraction sheet developed to extract relevant information for further analysis (Table 1).

Table 1 Data extraction sheet.

A data extraction sheet was developed to extract relevant information needed for further analysis and create themes for this study.

N	Study information	Objective	Participants	Study design	Results	
1	McKernan et al., 2022. USA	To reduce LUTS in a sample of individuals with chronic pain	64 adults with chronic pain and LUTS	A qualitative descriptive study	Hypnosis has the potential to drastically improve LUTS in individuals with chronic pain.	
2	Schroeder et al., 2021. Camden, NJ	To determine the differences between traditional conversation-based patient counseling and multimedia-based patient counseling	98 OAB patients	A randomized controlled trial	Multimedia-based patient education represents an effective method of providing patient education.	
3	Gulsen & Beji, 2022. Turkey	To determine the effect of healthy lifestyle behavior training, based on the Health Promotion Model (HPM), on the treatment of women with Overactive Bladder (OAB)	100 women diagnosed with OAB	A randomized controlled trial	HLSB training reduces OAB symptoms among women, improves the quality of their lives, changed their HLSBs, and positively affects their psychological symptoms.	
4	Funada et al., 2020. Japan	To evaluate the efficacy of CBT for OAB	150 OAB patients	A randomized controlled trial	The effectiveness of CBT with a structured manual may not only lead to a new treatment option for patients suffering from OAB symptoms, but may also reduce the social burden by OAB.	
5	Funada et al., 2020. Japan	To evaluate the feasibility and acceptability for drug-resistant OAB in women of the treatment model which was developed based on cognitive behavioral therapy	10 women with drug-resistant OAB	A single-arm pilot study	The new treatment based on cognitive behavioral therapy was well tolerated and feasible in women with drug-resistant OAB	
6	Svihra, 2020. Slovak Republic	To evaluate the efficacy and side effects of combined therapy with duloxetine and PFMT and duloxetine treatment alone	158 probands	A randomized intervention, parallel, multicenter study	-provide evidence of the efficacy of this combined treatment for SUI and highlight benefits associated with active approaches to treatment through exercise.	
7	Kosilov et al., 2019. Russian Federation	To study the effect of socioeconomic status (SES) on health-related quality of life (HRQoL) among persons with overactive bladder (OAB)	1,893 OAB patients	Survey	The improvement of HRQoL in persons with OAB is contingent upon increment in their level of awareness about the methods of OAB treatment and the effectiveness of treatment for severe symptoms of LUT pathology, increased social support and, possibly, physical activity.	
8	Xu et al., 2018. China	To investigate whether an education program targeting toileting behaviors is effective for helping overactive bladder patients with type 2 diabetes in terms of adopting healthy toileting behaviors, improving bladder symptoms, and enhancing quality of life	104 T2DM with OAB	A randomized controlled trial	Among overactive bladder patients with type 2 diabetes, the 6-week education program targeting toileting behaviors resulted in the adoption of healthy toileting behaviors, relief of bladder symptoms and improvement in quality of life in the 6 months following the intervention compared with routine care alone.	
9	Gezginci, Emine & Yilmaz, 2018. Turkey	to compare the effect of 3 instructional methods for behavioral therapy on lower urinary tract symptom (LUTS) severity and health-related quality of life (HRQOL) in women with OAB	60 women diagnosed with OAB	A randomized controlled trial	Structured verbal instruction plus educational leaflet is the most effective method of bladder training in women with overactive bladder and urge UI.	
10	Andrade et al., 2015. Miami, FL	To determine whether an avatar-based, online, self-management program is an effective therapeutic approach for women with overactive bladder (OAB).	41 female patients with OAB	A Randomized Controlled Trial	An avatar-based intervention embedded into an online self-management program improved OAB HRQOL and symptoms in women.	
11	Wang et al., 2011. China	To evaluate the prevalence, associated risk factors and the impact on HRQoL of OAB in China (age ≥18 years)	21,513 individuals	Survey	The symptoms of OAB have a detrimental effect on HRQoL. Efforts need to be made to improve public and professional education about the problems of OAB.	
12	Michel et al., 2011. Netherlands	To compare multiple single-item scales at baseline and after treatment with patient-reported overall rating of treatment efficacy	4,450	Survey	The VAS and the bladder problem question of the KHQ show the greatest promise as single-item scales to assess problem intensity in OAB patients.	
13	Zimmern et al., 2010. USA	To explore whether instruction in fluid management resulted in changes in fluid intake and incontinence over a 10-week study period in women with UUI	307 women with UUI	A Randomized Controlled Trial	General fluid instructions can contribute to the reduction in UUI symptoms, but additional individualized instructions along with other behavioral therapies did little to further improve the outcome.	
14	Herschorn et al., 2004. Canada	To assess a standardized and simple educational intervention in OAB patients to improve compliance with anticholinergic medication, increase the use of concomitant behavioral treatments, and improve patients’ perception of bladder symptoms	138 OAB patients	A Randomized Controlled Trial	The simple education intervention resulted in a greater, but not significant, increase in compliance with medication compared to the control group. It also resulted in a significantly increased use of behavior modification therapies and better self-perception of treatment outcome	
15	Vaughan et al., 2019. Alabama	Determine the efficacy of behavioral therapy for urinary symptoms in Parkinson’s disease.	53 patients diagnosed with Parkinson and OAB	A randomized clinical trial	Self-monitoring resulted in fewer urinary symptoms; however, only multicomponent behavioral therapy was associated with reduced bother and improved quality of life.	
16	Chu et al., 2016. USA	To determine if treatment of OAB can improve self-reported limitations in physical activity in women.	137 women with OAB	A prospective study	Treatment of OAB is associated with a decrease in perceived physical activity limitations but not directly associated with improvement in urinary symptoms.	
17	Majumdar et al., 2010. UK	To assess the effectiveness of inpatient bladder retraining	114 inpatients	an observational study by retrospective case-note analysis	The study confirms the usefulness of inpatient bladder retraining as a treatment option, especially in people refractory to outpatient management.	
18	Firinci et al., 2020. Turkey	To evaluate the efficacy of single and combined use of biofeedback (BF) and electrical stimulation (ES) added to bladder training (BT) on incontinence-related QoL and clinical parameters in women with idiopathic OAB	70 women with OAB	A prospective randomized controlled trial	In the first-line conservative treatment of women with idiopathic OAB: (i) adding BF and/or ES to BT increases treatment effectiveness, (ii) clinical efficiency is greater when the combination includes ES, (iii) BT + BF + ES (triple combination) is the most effective treatment option in reducing nocturia and improving QoL.	
19	Sung et al., 2015. Korea	to determine whether a health education intervention (HEI) could improve drug persistence with anticholinergics in OAB patients.	682 OAB patients	A randomised, open-label, multi-center trial	The health education intervention was not effective to increase drug persistence in OAB patients on anticholinergics	
20	Morris, Westbrook & Moore, 2008. Australia	To evaluate the long-term clinical outcome in women with idiopathic detrusor overactivity (IDO) and to identify significant prognostic factors.	132 women with detrusor overactivity	A longitudinal study	IDO seldom resolves and fluctuates in severity. Individual response is unpredictable, although the presence of urge incontinence is associated with a significantly worse prognosis.	
21	Firoozi et al., 2013. USA	We assessed how a group shared appointment influenced patient preparedness for sacral nerve stimulation for refractory overactive bladder and/or urge urinary incontinence. We also evaluated subjective and objective outcomes.	36 women with refractory OAB and/or UUI	A Randomized Controlled Trial	Participating in a group shared appointment before sacral nerve stimulation improved patient preparedness and perceived outcomes of treatment, although there was no difference in objective outcomes based on voiding diary.	

Progress in OAB psychological intervention

The psychological etiology of patients with OAB involves psychological and social factors. Therefore, we should attach importance to psychological intervention in treatment. Psychological intervention techniques which commonly used include health education, behavior therapy, cognitive behavioral therapy, group counseling, acceptance and commitment therapy and mindfulness intervention, et al. (Table 2).

Table 2 Psychological approaches to OAB patients.

Psychological approach	Objective	
Health education	To help patients understand treatment options and improve patients’ treatment compliance.	
Behavioral therapy	To make patients perform bladder training through the combination of specific behavioral training.	
Cognitive Behavioral Therapy	To treat OAB by changing the course of cognitive distortion in patients with OAB.	
Others	To cure OAB by changing factors in the course of OAB psychopathological process of OAB through psychological technique.	

Health education

The basis of disease treatment and nursing is chronic health education. Studies have found that the education level of patients can help patients understand treatment options and improve patients’ treatment compliance (Kosilov et al., 2019; Herschorn et al., 2004). Discussion and evaluation of OAB patients by general practitioners and physiotherapists can improve patient compliance to some extent (Wang et al., 2011), and thus improving patients’ quality of life (Xu et al., 2018), but health education alone does not improve patient outcomes in the long run (Liu, Jin & Zhang, 2019). In addition, behavior modification therapy was used more frequently and the self-perception of progress was improved (Herschorn et al., 2004). However, health education interventions do not enhance medication compliance or persistence in patients with OAB, according to the study of Sung et al. (2015).

With the development of science and technology and the acceleration of the pace of life, the forms of health education are gradually enriched. The study of Schroeder et al. (2021) found that multimedia-based patient education is an effective method of educating patients about urinary incontinence because those who received video education showed similar results as those who received standard physician counseling. In a urogynecology setting, video education can potentially replace or complement current patient education practices. There is evidence that the online OAB self-management program (Andrade et al., 2015) can help improve clinical symptoms and health-related quality of life in patients with OAB, providing a new approach to chronic management of overactive bladder.

Behavioral therapy

Based on health education, behavioral therapy makes patients correctly understand the bladder physiological function, and then realize the wrong urination habits they have and consciously change through health education (Herschorn et al., 2004). Through various forms of learning, patients can master better self-control ability and achieve good control of visceral physiological activities, behaviors and emotions (Chu et al., 2016). In the treatment of OAB, behavioral therapy is often used before or in combination with drug therapy and belongs to the first-line therapy. Behavioral therapy is widely used in clinical adjuvant therapy for OAB because of its characteristics like simplicity, no adverse reactions and no application authority.

The healthy lifestyle behaviors training based on health promotion model reduced OAB symptoms among women, improved their quality of lives, changed their HLSBs, and positively affected their psychological symptoms (Gulsen & Beji , 2022). Most patients diagnosed with OAB are treated as outpatients, while more severe cases require inpatient treatment. Inpatient bladder retraining has been found to be effective, especially for those who are unable to manage their condition outpatiently (Majumdar et al., 2010). Not only can behavioral therapy relieve clinical symptoms in patients with OAB, but also urinary symptoms in Parkinson’s patients (Vaughan et al., 2019). In the course of behavioral training, the Visual Analog Scale (VAS) and King’s Health questionnaire (KHQ) can be used to evaluate the effects of training (Michel et al., 2011). With a urination diary, the patient’s fluid intake, the amount of urination and the incidence of incontinence can be accurately tracked within 72 h. As a self-management tool, it is important for patients with OAB (Firoozi et al., 2013).

One treatment plan may not be suitable for all patients due to the obvious differences in individual patients, so targeted behavioral treatment plans should be developed according to patients’ conditions (Morris, Westbrook & Moore, 2008). Among women with OAB undergoing conservative first-line treatment, behavioral therapy in combination with other therapies, such as biofeedback and electrical stimulation, increases treatment effectiveness and improves quality of life (Firinci et al., 2020). According to Zimmern et al. (2010), general fluid instructions may help reduce UUI symptoms in women, but additional individualized instructions combined with other behavioural therapies had little effect on the outcome. Other studies have shown that active approaches such as pelvic floor muscle training can benefit patients and provide an innovative approach to female stress urinary incontinence (Svihra, 2020).

Methods of bladder training: The patients are instructed to urinate once at an interval of 30 min to 1 h, and the urination interval is extended 15 to 30 min every week until the urination interval is extended to 3 to 4 h.

Methods of pelvic floor muscle training: Take any position after the patient calms down, alternately perform the contraction and relaxation of urethra, perineum and anus, while keeping the buttocks, abdomen and femoral muscles in a relaxed state, and the contraction and relaxation movements were more than 5s. As the mastering of the key points of the motion, patients should practice 15 times each in the morning, middle and evening according to the pelvic floor muscle training method, which lasts for at least 10 weeks and may achieve satisfactory results.

Cognitive behavioral therapy

OAB can be treated effectively with pharmacotherapy, but there are risks associated with it as well. Behavioral therapy is another effective treatment for OAB; however, there are very few structured treatment manuals about how to prescribe it. Cognitive behavioral therapy (CBT) involves structured sessions involving therapists and patients collaborating on empiricism to solve problems. Symptoms of OAB are expected to worsen as cognitive distortion occurs. Studies have shown that CBT with a structured manual may not only result in a new treatment option for OAB patients, but may also reduce the social burden attached to the condition (Funada et al., 2021; Funada et al., 2020; Gezginci, Emine & Yilmaz, 2018). Figure 2 shows that CBT techniques are applied to OAB and explained in detail (Funada et al., 2020). It is expected to effectively treat OAB by changing the course of cognitive distortion in patients with OAB (Funada et al., 2020).

Figure 2 CBT model for OAB and target of each technique.

Overactive bladder treatment methods based on cognitive behavioral therapy (Funada et al., 2020): The training program consisted of four sessions (30 min each) and 6 techniques that were done over a period of 4 to 12 weeks. In each session, homework is reviewed, new techniques are presented, and the current day’s lesson is discussed. All sessions will be held individually, face-to-face. (1) For self-monitoring purposes, participants compile frequency volume charts during the intervention period. (2) The educational process involves teaching participants about the normal urinary tract system, abnormal voiding function, and the epidemiology and physiology of OAB. Our study applied the CBT model to OAB and explained the CBT technique’s targets. (3) Participants’ habits are assessed in order to modify their lifestyles, such as restricting their consumption of water and coffee if necessary. (4) Training of the pelvic floor muscles is carried out using abdominal breathing and a yoga ball to make it easier for the student to visualize. (5) Exposure is modified based on bladder training techniques in order to treat OAB. (6) Prevention of relapse involves planning for the future and encouraging the continuation of the techniques that have been taught. homework is assigned weekly in order to assess their adherence to the techniques.

Others

In addition to cognitive behavioral therapy, there are many schools of psychology, each of which has relatively mature psychological treatment programs, so it is possible to examine the impact of psychological treatment on OAB patients’ quality of life. The study conducted by McKernan et al. (2022) suggests that it is possible to dramatically improve lower urinary tract symptoms through hypnosis, even when no pain reduction occurs. For definitive results of hypnosis, randomized trials are required.

Apart from the above, different schools of psychology have options like psychoanalysis, Gestalt, Acceptance Commitment Therapy(ACT), Group Counseling, and Solution-Focused Brief Therapy. There is few research about these options in relevant database we have researched. Researchers need to work together to apply psychological treatment to chronic disease management of OAB. However, it should be noted that psychological intervention should be carried out in the disease after treatment symptoms are stable, or at the same time of clinical treatment, to obtain the maximum effect.

Discussion

Findings

To sum up, as a psychosomatic disease, OAB is closely related to psychological and social factors and can affect many aspects such as life, work, study and social activities. Mental health remains pivotal worldwide (Gao et al., 2021; He et al., 2021a; He et al., 2021b). In today’s society, the pace of life is speeding up with fierce competition, and people’s life is becoming more and more stressful, in which many people feel physically and mentally exhausted. However, in the investigation of morbidity, OAB has the characteristics of high morbidity, low hospitalization rate and high recurrence rate, having a serious impact on the quality of life of patients (Xin et al., 2019; Hu et al., 2018).

In the treatment process of OAB, despite clinical treatments, intensive guidance should also be considered from the aspects of patient health education, behavioral therapy and psychotherapy. Targeted health education can improve patients’ cognition of OAB, improve their awareness rate, cure rate and control rate of OAB, guide patients to face and solve problems in a positive way, and reduce the recurrence rate of OAB. At the same time, the severity of OAB symptoms is closely related to psychological factors. Medical personnel should pay attention to the psychological and social functions of patients. They can reduce the impact of OAB symptoms on patients’ lives by adjusting their lifestyle and behavioral habits (such as drinking more water during the day and drinking less water after dinner, etc.), so as to provide patients with a better quality of life.

Psychotherapy is woven into health education and behavioral therapy. Through the use of psychological counseling techniques of various psychological counseling schools (psychoanalysis, cognitive behavior, humanism, family therapy, gestalt, etc.) to improve patients’ cognition of OAB, improve patients’ psychological endurance, establish and cultivate patients’ good mental resilience, improve patients’ sense of self-efficacy and help patients establish confidence in rehabilitation, improve patients’ quality of life. In addition, psychological treatment can also explore the patient’s personality characteristics, character characteristics and growth experience, solve the patient’s psychological confusion, and then improve the patient’s quality of life and contribute to the patient’s recovery.

Further perspectives

In view of the fact that currently there is only a questionnaire OABSS on the market for evaluating overactive bladder patients’ severity, and the psychological evaluation of patients with OAB can only rely on the questionnaires of various psychological dimensions, and each dimension requires a scale, which is too cumbersome, so the scale can be prepared for comprehensive evaluation of clinical symptoms and mental health status of patients with OAB.

In addition, there are many schools of psychology, each of which has relatively mature psychological treatment programs, so it is possible to explore the intervention of psychological treatment programs on the quality of life of OAB patients.

Conclusion

As a psychosomatic disease, OAB is closely related to psychological and social factors and can affect many aspects such as life, work, study and social activities. Medical personnel should pay attention to the psychological and social functions of patients. In the treatment process of OAB, despite clinical treatments, intensive guidance should also be considered from the aspects of patient health education, behavioral therapy and psychotherapy. Psychotherapy is woven into health education and behavioral therapy. To explore and adjust the overall psychosomatic state of patients through psychological means, we may achieve the goal of promoting the rehabilitation of patients and improving the quality of life of patients.

Limitations

At present, there is only OABSS (Homma et al., 2006), a questionnaire used to evaluate the severity of overactive bladder, but no scale specifically used to evaluate the psychological dimension of OAB. Psychological problems of OAB require psychological questionnaires, such as psychological resilience scale, self-efficacy questionnaire, self-esteem scale, HADS, etc., so it is not described in this paper. In addition, medication to improve mental state is not covered in this article.

We thank the Home for Researcher for the linguistic editing and proofreading of the manuscript.

Additional Information and Declarations

Competing Interests

Author Contributions

Data Availability

Tian Li is an Academic Editor for PeerJ.

Zhaofeng Jin performed the experiments, authored or reviewed drafts of the article, and approved the final draft.

Qiumin Zhang performed the experiments, authored or reviewed drafts of the article, and approved the final draft.

Yanlan Yu performed the experiments, authored or reviewed drafts of the article, and approved the final draft.

Ruilin Zhang analyzed the data, prepared figures and/or tables, and approved the final draft.

Guoqing Ding analyzed the data, prepared figures and/or tables, and approved the final draft.

Tian Li conceived and designed the experiments, authored or reviewed drafts of the article, and approved the final draft.

Yuping Song conceived and designed the experiments, analyzed the data, authored or reviewed drafts of the article, and approved the final draft.

The following information was supplied regarding data availability:

This is a review article and did not generate raw data.

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
