# Peer review of "Progress in overactive bladder: novel avenues from psychology to clinical opinions"

_PeerJ, doi:10.7717/peerj.16112_

## Round 0.1 · original submission · Major Revisions

This manuscript lacks a discussion of behavior and psychotherapy, and no review algorithm is provided. In addition, the article is slightly lengthy, so it is recommended to simplify the manuscript.

·

Basic reporting

In the article: Progress in overactive bladder: Novel avenues from psychology to clinical opinions authors approached the aspects of non-surgical treatment for OAB. Despite the name: “Novel avenues from psychology to clinical opinions”, the discussion was mainly regarding general characteristics of OAB and lesser regarding behavioral treatments, which described widely in the literature. The article is very long, describing detailly the epidemiology, etiology and consequences of the problem, but give the little attention to behavioral treatment and even less for psychologic approach.
Phrasing should be repaired and English language should be revised.

Experimental design

The review is not systematic, the literature searching algorithm is not provided

Validity of the findings

Psychologic treatment for chronic illness, in this article OAB is very important issue. Unfortunately, the article didn't provide any information regarding the actual strategies of treatment, description of treatment types was unclear.
I suggest for the authors to consider significant condensation of the article, especially epidemiology, etiology and consequences parts, as nothing new is reported in those portions.
To describe in details the psychologic aspects of the disease and types of psychological treatments for OAB, which can be applied in the outpatient urogynecological clinics. Provide the charts with psychological approach to chronic patient and particularly to OAB patient.

Additional comments

There are few remarks for repair:
When you are reporting the epidemiology data, you have to describe the gender differences: apart of the fact that females have more Urge incontinence they have significantly more OAB.
The epidemiology part is very long, condense the data.
Line87 When you describe the consequences of OAB in details, you can’t wright “and others), if any others you should add it.
Line 92 Lack of hospitalizations with OAB diagnosis doesn’t mean that patients don’t look for help in outpatient clinics, OAB itself is not indication for hospitalization.
Line 242- Add reference
The figures don’t provide any information, for my opinion.

Reviewer 2 ·

Basic reporting

The paper reviewed the progress of psycholog on OAB from three aspects: epidemiology, psychological etiology, and psychological intervention. However, a revelation from the analysis of three aspects is that OAB is indeed a systemic disease, not just a psychiatric or urinary-related.

Experimental design

The overall design idea is good, and the main focus is on the psycho- aspects and analysis of OAB in the urinary system. For example, according to the psychopathological network theory, the Figure 3 says that stress is the starting factor at the beginning, and then it will affect the relationship between the two. At this time, if the stress is no longer there, but the two still interact with each other and worsen, however, this is a difficult thing to define. Urinary symptoms and psychiatric factors may exacerbate each other, but to OAB, it shouldn't be an antagonistic relationship.

Validity of the findings

There is no definitive conclusion to this three-pronged statement, which tells us why OAB is related to psycho-realated things. What is worth thinking about in between.

Additional comments

OAB is just a symptom complex in urological system, and psychiatric disorder is also a symptom complex related. There may be a connection between them. Maybe OAB is just a manifestation in urinary system after some factors affect the whole body systems. And the distribution of the network also involves the mental- psyco- factors, there are also other inoloved related systems.

---

## Round 0.2 · Minor Revisions

Please shorten the “Scope” portion in the first introduction. In addition, please improve the discussion section according to the reviewer's suggestions.

·

Basic reporting

I would recommend to delete or significantly shorten the “Scope” portion in the first introduction.

Experimental design

Correct

Validity of the findings

Important for education of practitioners

Additional comments

Dear Editor and Dear authors.
There is a significant improvement in your work after the revision.
I would like to make some additional remarks, for my opinion after minor corrections the article may by published.
I would recommend to delete or significantly shorten the “Scope” portion in the first introduction.
To the audience to add the general practitioners and physiotherapy practitioners. When we are talking about poor compliance the general practitioner may initiate the discussion and evaluation of the patient urinary complains during a regular visit. It may be very helpful in the diagnosis of a new cases and compliance improvement.
79 The low hospitalization rate doesn’t mean that patients do not choose to seek care. There is no need for hospitalization, but for outpatient follow up. Please paraphrase.
233 Please, explain the abbreviations: HLSB training based on HPM
285 LUTS- abbreviation
294 Discussion- Start with description of your findings, discuss the findings via your perspective, comparing to others, at the end add the limitations, it shouldn’t be at the beginning of the discussion part.
312 Conclusion part should contain several sentences only. I suggest to transfer this part to discussion and to add real short conclusion at the end.

Reviewer 2 ·

Basic reporting

The authors completed the required revisions. No more comments.

Experimental design

OK

Validity of the findings

No comment

Additional comments

OK

---

## Round 0.3 · accepted · Accept

The authors have addressed the questions and the paper may be accepted.